# The SWGEDWGEIW from Soybean Peptides Reduce Oxidative Damage-Mediated Apoptosis in PC-12 Cells by Activating SIRT3/FOXO3a Signaling Pathway

**DOI:** 10.3390/molecules27217610

**Published:** 2022-11-06

**Authors:** Guofu Yi, Mengyue Zhou, Qingfei Du, Shuibing Yang, Yuxia Zhu, Yining Dong, Yang Liu, He Li, You Li, Xinqi Liu

**Affiliations:** 1College of Biology and Food Engineering, Chuzhou University, Chuzhou 239000, China; 2Beijing Advanced Innovation Center for Food Nutrition and Human Health, Beijing Engineering and Technology Research Center of Food Additives, Beijing Technology and Business University (BTBU), Beijing 100048, China

**Keywords:** SWGEDWGEIW, soybean peptides, PC-12 cells, mitochondrial dysfunction, neuronal oxidative damage

## Abstract

The goal of the investigation was to study the protective effects of the SWGEDWGEIW (the single peptide, TSP) from soybean peptides (SBP) on hydrogen peroxide (H_2_O_2_)-induced apoptosis together with mitochondrial dysfunction in PC-12 cells and their possible implications to protection mechanism. Meanwhile, the SBP was used as a control experiment. The results suggested that SBP and TSP significantly (*p* < 0.05) inhibited cellular oxidative damage and ROS-mediated apoptosis. In addition, SBP and TSP also enhanced multiple mitochondrial biological activities, decreased mitochondrial ROS levels, amplified mitochondrial respiration, increased cellular maximal respiration, spare respiration capacity, and ATP production. In addition, SBP and TSP significantly (*p* < 0.05) raised the SIRT3 protein expression and the downstream functional gene FOXO3a. In the above activity tests, the activity of TSP was slightly higher than that of SBP. Taken together, our findings suggested that SBP and TSP can be used as promising nutrients for oxidative damage reduction in neurons, and TSP is more effective than SBP. Therefore, TSP has the potential to replace SBP and reduce neuronal oxidative damage.

## 1. Introduction

One of the chronic and life-threatening conditions is a neurological disorder, which negatively affects people’s well-being [1] with only a few intervention avenues [2]. Because of the growing and aging population recently, the prevalence of diseases has been constantly increasing. Thus became the top reason for disability and the second leading cause of death globally [3]. Oxidative stress and mitochondrial dysfunction are usually the causes of neurodegenerative diseases such as Alzheimer’s disease (AD), which are mainly manifested as emotional and cognitive changes, autonomic dysfunction, and sleep disorders [4]. A growing body of research suggests that sleep disturbances lead to decreased memory function and cognitive impairment [5]. There is a bidirectional relationship between sleep and Alzheimer’s disease, who influence each other [6]. Studies suggest that systemic inflammation caused by sleep disturbance raises the chance of AD and is regarded as the development of AD’s driving factor [7]. Interestingly, the risk of AD could also be reduced by good sleep [8]. It is generally accepted that diet deeply affects the occurrence as well as the development of neurodegenerative diseases [9,10,11]. Foods rich in tryptophan not only promote sleep but also prevent neurodegenerative diseases [12]. Soybean protein is generally known as full-value protein, rich in tryptophan as well as soybean peptides (SBP) prepared by enzymatic hydrolysis that can resist oxidative stress, promote sleep, be free of allergies, and can be widely consumed [13,14,15,16,17,18]. Moreover, compared to drug prevention, SBPs are safe and have no side effects as a food source. However, various adverse effects are associated with drug therapy, such as mitochondrial toxicity, hepatotoxicity, metabolic/endocrine adverse events, and neurological toxicity [19,20]. Jiayan Ren’s research on antioxidation and anti-aging of protease hydrolyzed peptides of soybean, oyster, and sea cucumber discovered that SBPs had the best effect, and two single peptides Trp-Pro-Lys and Ala-Tyr-Leu His were obtained from them. These two peptides can significantly reduce H_2_O_2_ induced oxidative PC12 cell damage. The results showed that SBPs could potentially be used to treat aging related issues and memory disorders [21]. Jing Wang et al. studied Piceatannol associate ROS-mediated PC-12 cells damage and mitochondrial dysfunction through Sirtuin 3(SIRT3)/Forkhead box O3(FOXO3a) signaling path, suggesting that SIRT3/FOXO3a signaling path played a vital role in mediating the neural cytoprotective effects of piceatannol [22]. SIRT3 is involved in neuroprotective activities, as evidenced by the growing body of research. SIRT3, a mitochondria-localized NAD+-dependent deacetylase protein, regulates stress resistance and energy metabolism [23], which is proven to be closely related to neuroprotective activity [24]. For example, SIRT3 ameliorates ApoE4-related memory and learning impairments as well as prevents neuronal damage and mitochondrial dysfunction in a deacetylase-dependent manner induced by P53 [25].

However, there is a lack of information on the influence of SBPs on oxidative damage to nerve cells and their neuroprotective mechanism, especially about SBPs involvement in SIRT3-mediated neuroprotection. According to our previous study, the single peptides (SWGEDWGEIW, TSP) with higher content and containing tryptophan were identified by Nano-LC/MS-MS from SBP [14].

PC-12 are nerve cells derived from adrenal medullary pheochromocytoma in rats. When cultured in vitro, it can show some growth characteristics similar to neurons, such as cell aggregation and fiber ridge emergency [26]. Therefore, the cells show many useful characteristics for neurotoxicity, neuroprotection, and neurocognitive research. H_2_O_2_ is one of the known major reactive oxygen species, which can increase oxidative stress and lead to apoptosis or necrosis of PC-12 cells, characterized by chromatin margination or condensation, DNA fragmentation, and nuclear collapse [27].

In this study, the (single peptide, TSP) current work aimed to explore the neuroprotective effects of SBP and TSP on mitochondrial dysfunction and PC-12 cell apoptosis induced by H_2_O_2_ as well as the underlying possible mechanisms to verify whether TSP can represent SBP-mediated SIRT3-mediated neuroprotection. Furthermore, the ability of SBP and TSP to inhibit H_2_O_2_-induced mitochondria-mediated apoptosis and cellular oxidative damage was also validated through FOXO3a/SIRT3 signaling pathway. Our investigation provided some insights into the extensive use of SBP and TSP as food in the prevention and treatment of neuronal degenerative diseases, which are expected to replace SBP with TSP.

## 2. Results and Discussion

### 2.1. Protective Effects on PC-12 Cells Oxidative Damage Induced by H_2_O_2_

Cytotoxicity of PC-12 cell growth was assessed by the CCK8 method. Treatment of H_2_O_2_ with concentrations higher than 250 μM for 24 h resulted in significant neurotoxicity in PC-12 cells (Figure 1A), while the presence of 250 μM and 500 μM H_2_O_2_ induced 46.18% and 77.82% cell death, respectively. Meanwhile, the reactive oxygen species increased with the elevation of the concentration of H_2_O_2_, especially when the concentration of H_2_O_2_ was 250 μM, the concentration of reactive oxygen species increased by nearly 3.67 times compared with the control group (Figure 1B). Therefore, selecting 250 μM H_2_O_2_ can meet the requirements of oxidative damage and can be used for subsequent experiments. When SBP or TSP was added at concentrations below 100 μg/mL, there was no significant toxicity, but when the concentration was added to 200 μg/mL, PC-12 cell viability was adversely affected (Figure 1C). Since two samples of SBP and TSP needed to be compared at the same time, 10 μg/mL and 100 μg/mL were selected for subsequent experiments.

As shown in Figure 1D, when SBP and TSP were treated with 250 μM H_2_O_2_-induced oxidative damage, they exhibited significant cytoprotective effects. With the prolongation of time, the oxidative damage continued to increase and did not change after 6 h, but the protective effect of SBP and TSP on cells was the best at 6 h. At the same time, TSP was more effective than SBP at the same dose. These results suggest that SBP and TSP can ameliorate PC-12 cells’ oxidative damage.

### 2.2. TSP and SBP’s Inhibition on PC-12 Cells’ Oxidative Stress-Induced Mitochondria-Dependent Apoptosis

Oxidative stress breaks mitochondrial function, gives out pro-apoptotic proteins, increases caspase-3 activity, leads to mitochondrial membrane permeability changes, and ultimately activates apoptosis [28]. As shown in Figure 2A, H_2_O_2_ stimulated mitochondrial ROS production, which was counteracted by SBP and TSP pretreatment. In particular, when TSP concentration was 100 μg/mL, compared with the positive control group, ROS concentration decreased by 3.3 times, while 100 μg/mL SBP concentration was decreased by 1.9 times, compared with the positive control group. Oxidative stress could cause apoptosis through mitochondria-independent and mitochondria-dependent pathways, as previously reported [29], the latter of which is characterized by the damage to the outer membrane of mitochondria followed by the cytochrome c release into the cytoplasm for caspases activation [30]. In this study, caspase-3 downstream expression was significantly inhibited by SBP and TSP treatment, which was cleaved via the mitochondria-dependent pathway, (Figure 2B). When the concentration of TSP and SBP was 100 μg/mL, it was nearly twice as much as that of the positive control group. Pro-apoptotic pro-cell-Teins Bak and Bax are transferred from the cytoplasm to the mitochondrial outer membrane during its apoptosis, inducing mitochondrial rupture, forming permeability transition pores, and allowing the release of apoptotic factors [30]. In contrast, the anti-apoptotic proteins Bcl-2 and Bcl-xL inhibited Bak and Bax activation and maintained the integrity of mitochondria. The data displayed that the expression of Bcl-2 protein in the SBP and TSP treatment groups was significantly increased compared with the H_2_O_2_ group, and the Bax protein expression was significantly inhibited in a dose-dependent regime. When the concentration of TSP and SBP was 100 μg/mL, compared with the positive control group, the value of Bcl-2/Bax increased nearly 3.2 times and 2.4 times, respectively, suggesting that SBP and TSP counteract H_2_O_2_-induced neurotoxicity by maintaining mitochondrial integrity and inhibiting mitochondria-dependent apoptosis.

### 2.3. SBP and TSP’s Protection against the Mitochondrial Respiratory Dysfunction Caused by Oxidative Stress

Mitochondria are the cardinal energy-producing organelles that power many key cellular activities. the effectiveness of maintaining the mitochondria’s activity can improve neuronal survival and limit oxidative stress [31]. In order to further explore the SBP protective effect on mitochondrial respiration, we measured PC-12 cells’ OCR values with different treatments. Rotenone, oligomycin, FCCP, and antibiotics were sequentially added as mitochondrial stressors, cells were detected and analyzed using the Seahorse XFe24 analyzer for respiratory capacities such as maximal respiration and ATP production, (Figure 3A,B). No significant change occurred in cellular basal respiration subject to H_2_O_2_ peroxide or SBP treatment (Figure 3C), but TSP did significantly change cellular basal respiration compared with H_2_O_2_ treatment. However, compared with the Blank control, ATP production and spared respiration volume were significantly decreased by H_2_O_2_ treatment, which were reversed by TSP treatment. However, SBP treatment did not cause any significant change (Figure 3D,E), which significantly enhanced the coupling efficiency of cells and maximal respiration, as well as counteracted the H_2_O_2_-induced mitochondrial respiratory dysfunction (Figure 3F,G). Furthermore, the H_2_O_2_-treated group significantly decreased the OCR/extracellular acidification rate (ECAR) compared with the Blank control. The H_2_O_2_-treated samples after pretreatment with SBP and TSP also displayed significant differences (Figure 3H). Taken together, these results suggest that SBP and SPEM preconditioning have protective effects on oxidative stress-induced mitochondrial respiratory dysfunction, with TSP being more effective than SBP.

### 2.4. SBP and TSP’s Prevention of Mitochondria-Mediated Apoptosis and Oxidative Damage through the Signaling Pathway of SIRT3-FOXO3a

In order to explore SBP and TSP’s molecular mechanism of protection against mitochondria-mediated apoptosis and oxidative damage, corresponding proteins’ expression was detected by Western blotting in the SIRT3-FOXO3a signaling pathway. As displayed in Figure 4A, compared with the Blank control, H_2_O_2_ exposure significantly curbed SIRT3 and FOXO3a levels, whereas SBP and TSP treatments reversed this trend and significantly upregulated FOXO3a and SIRT3 expression in a dose-dependent regime, TSP increased 30% and 10% in up-regulation of SIRT3 and FOXO3a compared to SBP. We also explored the role of SIRT3 in regulating H_2_O_2_-induced apoptosis and cell death by SBP and TSP. SBP and TSP-induced apoptosis were reduced compared with H_2_O_2_ treatment alone, suggesting that SBP attenuated H_2_O_2_-induced PC-12 cells’ mitochondria-dependent apoptosis (Figure 4B). These results indicate that SBP and TSP can reduce apoptosis and cell damage in the SIRT3-FOXO3a signal pathway. Mitochondrial function and protection from oxidation regulation have been shown to involve SIRT3, which can also regulate the redox state of cells by activating FOXO3a target genes and itself, thereby showing its antioxidant capacity [32] and preventing apoptosis by inhibiting mitochondrial dysregulation as well as reducing ROS production [33]. In neurons, ROS progressive accumulation leads to neuronal degeneration, abnormal protein aggregation, cell death, and mitochondrial dysfunction [34].

Based on the above results, we proposed a neuroprotective effect of SBP and TSP. This could be due to their antioxidant activity as well as the sleep-promoting effect that nourishes cranial nerves which is aligned with our previous result wherein HepG2 cells can be protected from oxidative damage via SBP [13]. In our previous work, it was found that increasing the ratio of tryptophan and melatonin in the mouse brain and controlling relevant metabolic enzymes enhanced the sleep-promoting activity of SBPs in mice. TSP contains a sufficient amount of tryptophan. The tryptophan ratio in TSP was nearly 25% higher than in SBP. Protein biosynthesis requires an essential amino acid of tryptophan, supplementing of which in the development of animal models of neurodegenerative diseases has shown promising results [12]. This might also be the cause of the effectiveness of TSP in this study over SBP. Furthermore, our current study demonstrated that SIRT3 inhibition may exacerbate apoptosis and speed up intracellular oxidative damage which is consistent with previous reports showing that knockdown of SIRT3 exacerbates mitochondrial dysfunction and neuronal apoptosis [35]. In light of this, the bioavailability of SBP and TSP in tissues (including the brain), plasma, and urine will therefore be studied in further research. Meanwhile, SBP and TSP protective mechanisms will be elucidated in clinical trials and in vivo.

## 3. Materials and Methods

### 3.1. Samples

SBPs were prepared according to research reports in our laboratory. In short, soybean protein is mainly obtained by enzymatic hydrolysis of various proteases [14]. As shown in Figure 5, the single peptide of 99% purity was synthesized and bought from GL Biochem Ltd., (Shanghai, China), TSP. The PC-12 cell line we used was purchased from Procell Life Science & Technology Co., Ltd., (Wuhan, China) Horse serum (HS), hank’s balanced salt solution (HBSS), and fetal bovine serum (FBS) were bought from Gibco Ltd., (Madison, WI, USA). Anti-SIRT3(ab217319), FOXO3A(ab109629), BCL2-associated X protein (Bax, ab32503), B-cell lymphoma-2(Bcl-2, ab32124), and Caspase-3 (ab32351) were purchased from Abcam Ltd., (Cambridge, UK). Anti-glyceraldehyde-3-phosphate dehydrogenase (GAPDH, ab181602) was provided by JK GREEN Ltd., (Beijing, China).

### 3.2. Rat Pheochromocytoma (PC-12) Culture

Cells were bought from Peking Union Medical College (Beijing, China), which were cultured with RPMI 1640 medium in Falcon culture dishes (Gibco, Grand Island, NY, USA) with 100 μg/mL streptomycin, 10% of HS, 5% of FBS, and 100 μg/mL penicillin in an incubator at 37 °C and filled with 5% CO_2_. The cells were separated at 80–90% confluency with 0.25% trypsin solution. Cell-counting assay kit (CCK-8) was purchased from Dojindo (Kumamoto, Japan).

### 3.3. Cell Viability Treatments Assessment

PC-12 cells were procured at 1 × 10^5^ cells/mL in 96-well plates (Corning, NY, America). An amount of 10 μL of free Dulbecco’s modification of Eagle’s medium Dulbecco (DMEM)with 10% of the FBS as control or FBS-free DMEM containing 0 μg/mL, 5 μg/mL, 10 μg/mL, 50 μg/mL, 100 μg/mL, 200 μg/mL and 400 μg/mL SBP or TSP and 0 μM, 15 μM, 30 μM, 60 μM, 120 μM, 250 μM, 500 μM, and 1000 μM H_2_O_2_ were added, and the cells were incubated for 30 min with 5% CO_2_ and 95% air at 37 °C. Cell viability assay was conducted according to CCK-8 kit (Dojindo, Japan) instructions [36]. The plate wells’ absorbance was recorded in an MK3 plate reader at 450 nm (Thermo Fisher Scientific, Waltham, MA, USA). 

### 3.4. Intracellular Apoptosis and ROS Detection

Intracellular ROS content was detected using the fluorescent probe DCFH-DA. PC-12 cells were placed in six-well plates with 1 × 10^5^ cells/mL. Sixteen to eighteen hours after adherent culture, cells are treated with polypeptides. After the treatment, cells were rinsed in HBSS 3 times, followed by incubation in a serum-free DMEM medium containing 20 μM DCFH-DA probe for 30 min. After incubation, the culture was mixed with pre-cooled DPBS three times and induced with H_2_O_2_ for half an hour in dark. A flow cytometer (CytoFLEXS, Beckman, CA, USA) was utilized to measure the fluorescence intensity. ROS assay kits were bought from the Beyotime Institute of Biotechnology (Shanghai, China). Apoptosis detection was performed with an Annexin V/PI apoptosis detection kit (BD, San Jose, CA, USA). PC-12 cells subject to different treatment groups were trypsinized and resuspended in 200 uL DPBS, then 5 uL Annexin V-FITC was added to each tube, mixed well, and the reaction was performed in the dark for 20 min. Then 5 uL of PI was added, and immediately after mixing, fluorescence detection was carried out by a flow cytometer (CytoFLEX S, Beckman, CA, USA).

### 3.5. Oxygen Consumption Rate (OCR) Determination

Plates loaded with Seahorse XF Calibrant (pH 7.4) were kept at 4 °C overnight. Intact cells’ OCR was determined with the Seahorse XFe24 Analyzer (Seahorse Bioscience, North Billerica, MA, USA). Each well of the 24-well XF Analyzer cell culture plates containing 1 × 10^5^ cells was incubated for 24 h. Cells were then seeded in 20 μM SBP or TSP for 48 h, followed by 40 min in 250 μM H_2_O_2_. Cells were rinsed 3 times, incubated at XF DMEM basal medium (pH 7.4) supplemented with 2 g/L of D-glucose and placed in a non-CO_2_ incubator at 37 °C before analysis. After 1 h of culture, OCR was measured under four different conditions. ATP synthase inhibitor was used by sequential injections of oligomycin (1 μM). The potent mitochondrial oxidative drug uncoupler was applied via FCCP (2 μM), mitochondrial electron transport chain complex I inhibitor was utilized by rotenone (1 μM), and potent NADH oxidation inhibitor. Spare respiratory capacity, basal respiration, ATP production, and maximal respiration were assayed for mitochondrial function by Seahorse Wave software 2.4 (http://www.seahorsebio.com (accessed on 5 December 2021)).

### 3.6. Western Blot Assay

Ice-cold RIPA lysis buffer was used to lyse the harvested PC-12 cells, which were incubated at 4 °C for 30 min and then centrifuged at 14,000× *g* at 4 °C for 15 min to procure the supernatant. The protein concentration of the supernatant was adjusted to 2 μg/μL with a loading amount of 4 μL after the BCA protein content. Sodium dodecyl sulphate-polyacrylamide gel electrophoresis (SDS-PAGE) was performed by adding 10 min-boiled supernatant together with the loading buffer [37], after which the sample was transferred to the Blank control membrane (Millipore, MA, USA) blocked for 1 h at room temperature by 5% skim milk, and incubated with the selected primary antibody overnight at 4 °C. TBST was used to rinse the membrane 5 times, and the secondary antibody was applied for 1 h at room temperature incubation. After washing, the proteins were determined with ECL (Millipore, MA, USA) with GAPDH as an internal reference.

### 3.7. Data Analysis

The results were expressed as the mean ± standard deviation (SD) with experiments being performed in triplicate. Different treatments were compared by SPSS (version 16.0, IBM Inc., Armonk, NY, USA) via one-way analysis of variance (ANOVA). *p* value less than 0.05 was set to be statistically significant.

## 4. Conclusions

In this study, biological techniques (OCR, ROS, WB, and so on) were used to assess the protective effects of SBP and TSP against oxidative stress in PC-12 cells. The results showed that SBP and TSP attenuated mitochondrial dysfunction and oxidative stress-induced cellular damage and suggested that activation of the SIRT3-FOXO3a signaling pathway exerted neuroprotective effects. In particular, the single peptide TSP was superior to SBP in terms of regulatory effect.

## Figures and Tables

**Figure 1 molecules-27-07610-f001:**
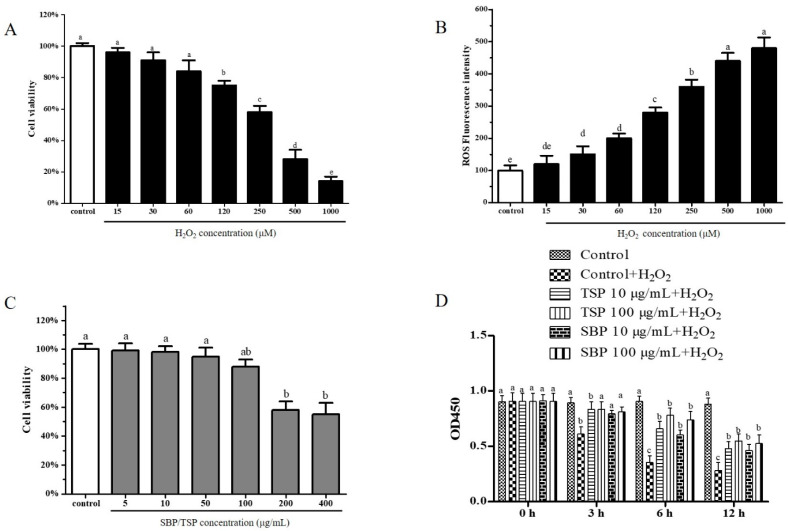
Viability of PC-12 cells under the influence of H_2_O_2_ (**A**), an PC-12 cells generation under the influence of H_2_O_2_ (**B**), Viability of PC-12 cells under the influence of peptides (**C**), PC-12 cells’ intracellular ROS scavenging capacities of different concentrations of peptides and different incubation time under oxidative stress induced by H_2_O_2_ (**D**). Data are expressed as the mean ± SD. The results marked with the different letters (a, b, c, d and e)are significantly different (*p* < 0.05).

**Figure 2 molecules-27-07610-f002:**
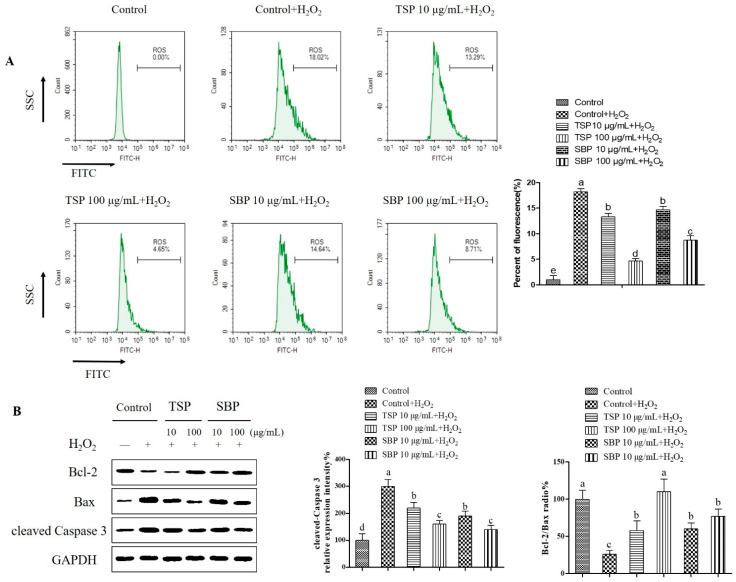
Protection of H_2_O_2_-treated PC-12 cells from mitochondrial-dependent apoptosis by SBP and TSP. 10 μg/mL and 100 μg/mL of SBP and TSP were pretreated to cells for 48 h followed by exposure to 250 μM H_2_O_2_. (**A**) SBP and TSP effect on mitochondrial ROS generation, which was detected by immunofluorescence. (**B**) SBP and TSP effect on protein expression. Cleaved-caspase 3, Bcl-2, and Bax protein expression levels were analyzed by Western blotting. Data are expressed as the means ± SD (*n* = 3), results marked with different letters (a, b, c, d and e) are significantly different (*p* < 0.05).

**Figure 3 molecules-27-07610-f003:**
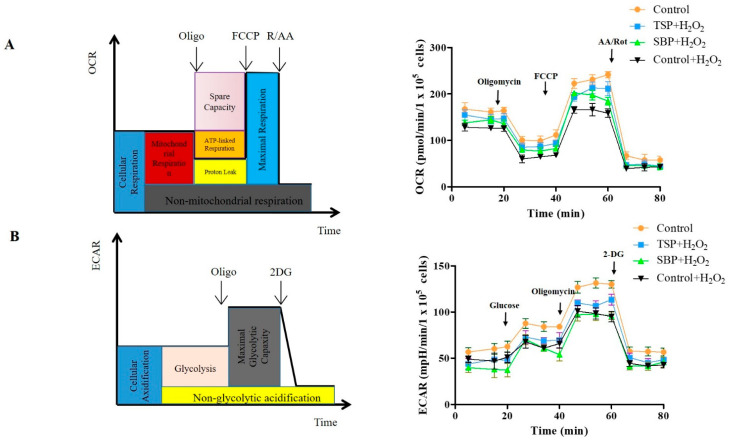
Improvements of SBP and TSP to H_2_O_2_-induced PC-12 cells’ mitochondrial dysfunction. 100 μg/mL of SBP and TSP was used to treat cells for 48 h and followed with exposure of 250 μM H_2_O_2_ for 40 min. Seahorse XFe24 Analyzer was the instrument to test OCR. (**A**,**B**) Mitochondrial respiration profile in cells with different treatments. (**C**) Basic respiration. (**D**) Production of ATP. (**E**) The capacity of spare respiratory. (**F**) Maximum respiration. (**G**) Coupling efficiency. (**H**) The ratio of OCR/ECAR. Data are expressed as the means ± SD (*n* = 3), results marked with different letters (a, b and c) are significantly different (*p* < 0.05).

**Figure 4 molecules-27-07610-f004:**
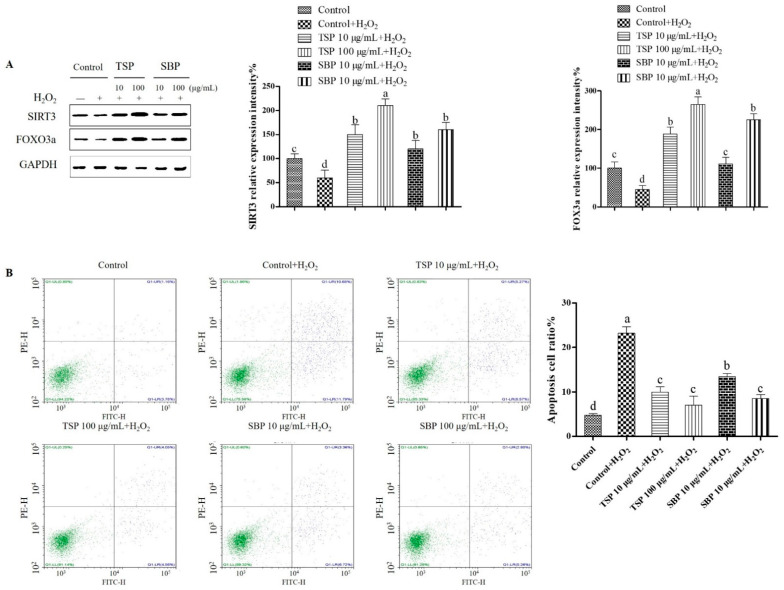
Prevention of mitochondria-mediated apoptosis and oxidative damage through the SIRT3-FOXO3a signaling pathway by SBP and TSP followed with cell incubation for 48 h and finally exposure to 250 μM H_2_O_2_. (**A**) SBP or TSP effect on SIRT3 and FOXO3a protein expression. (**B**) PC-12 cells’ mitochondria-dependent apoptosis induced by H_2_O_2_ in SBP and TSP attenuated manner. Data are expressed as the means ± SD (*n* = 3), results marked with different letters (a, b, c and d) are significantly different (*p* < 0.05).

**Figure 5 molecules-27-07610-f005:**
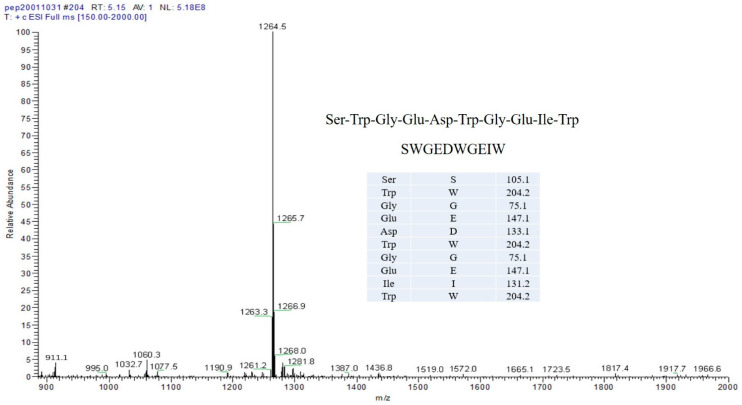
Mass spectra of TSP.

## Data Availability

The data presented in this study are available on request from the corresponding author.

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
