# Peer review of "The SWGEDWGEIW from Soybean Peptides Reduce Oxidative Damage-Mediated Apoptosis in PC-12 Cells by Activating SIRT3/FOXO3a Signaling Pathway"

_molecules, 2022, doi:10.3390/molecules27217610_

Round 1
Reviewer 1 Report
Comments: Minor revision
The manuscript on The SWGEDWGEIW from Soybean Peptides which Reduce 2 Oxidative Damage-mediated Apoptosis in Neurons and Mito- 3 chondria by Activating SIRT3/FOXO3a Signaling Pathway nicely discussed most of the critical issues in this field. The topic seems very interesting in the field since several controversies are present. Authors have tried their best to represent a comprehensive article is itself very interesting. Some sentences are too long to follow, please shorten them or rephrase them. Please avoid redefining the abbreviations throughout the text. I support publication of this review article major language modifications and spell check,
1. In the table, references are not properly cited. Authors are suggested to correct it as per journal guidelines.
- Elaborate introduction section.
- The abbreviations must be checked throughout the manuscript.
- The article must be checked for English proofing and grammatical errors.
- I could suggest making two different sections for eg. (Future perspectives and conclusion). It should be rewritten
6. Finally, the references must be formatted as per the journal requirement and several syntax errors exist within the manuscript, which must be corrected.
Author Response
Response to Reviewer 1 Comments
Dear honoured reviewers,
We really appreciate you and the reviewers for your comments on our Manuscript: “The SWGEDWGEIW from Soybean Peptides which Reduce Oxidative Damage-mediated Apoptosis in PC-12 cells and Mitochondria by Activating SIRT3/FOXO3a Signaling Pathway” We express our sincere gratitude and thankfulness for your time and precision in reviewing our manuscript. The responses to the comments are as follows. For your kind information, we have carefully dealt with the comments of the reviewers as follows: All the places which we changed have already been marked yellow in our paper, for the purpose of highlight. We hope the revised manuscript meets the standard of publication. Thank you!
Point 1: Please avoid redefining the abbreviations throughout the text. I support publication of this review article major language modifications and spell check.
Response 1: Thank you very much for the comments and suggestions. All the abbreviations have been checked and the redefined abbreviations have been removed. We have done the language editing and spell checking as you suggested. The whole manuscript has been modified.
Point 2: In the table, references are not properly cited. Authors are suggested to correct it as per journal guidelines.
Response 2: Thanks for your suggestion. We have made corrections to the reference citation according to the journal guidelines, and marked in yellow color from lines 330-412.
Point 3: Elaborate introduction section.
Response 3: Thanks for your suggestion. We have elaborated the introduction section from lines 50-76.
Point 4: The abbreviations must be checked throughout the manuscript.
Response 4: Thanks for your suggestion. The abbreviations have been checked and modified throughout the manuscript. We have added the abbreviations used in the entire manuscript to the final abbreviations section below conclusion (Lines 304-321).
Point 5: The article must be checked for English proofing and grammatical errors.
Response 5: Thanks for your suggestion. We have corrected the English language and grammar errors in the whole manuscript.
Point 6: I could suggest making two different sections for eg. (Future perspectives and conclusion). It should be rewritten
Response 6: Thanks for your suggestion. We have rewritten the conclusion for more clarity (lines 297-303).
Point 7: Finally, the references must be formatted as per the journal requirement and several syntax errors exist within the manuscript, which must be corrected.
Response 7: Thanks for your suggestion. We have formatted the references according to the journal format, and grammatical errors in the manuscript have been corrected.
Reviewer 2 Report
1. In the title of the manuscript it’s written “neurons “, but all the experiments have been done only on PC-12 cell line. There were not any neurones used as the object of research, because PC-12 cell line is pheochromocytoma cell line. So, to speculate that “TSP has the potential to replace SBP and reduce neuronal oxidative damage.” may be not correct.
2. The authors write in the “Conclusions” section: that “it is for the first time”… but there is some data on this topic. For example, where the soybean peptides were purified and analyzed in detail. [Amakye, W. K., Hou, C., Xie, L., Lin, X., Gou, N., Yuan, E., & Ren, J. (2021). Bioactive anti-aging agents and the identification of new anti-oxidant soybean peptides. Food Bioscience, 42, 101194. doi:10.1016/j.fbio.2021.101194]
3. Throughout the paper the authors single peptide (TSP) with the total soybean peptide. The difference is not too big, as it could be seen on the figures. There are not any digital presentation of the difference in percentiles, only visual. For example, as it is displayed in Figure 5A, compared with the NC group, hydrogen peroxide exposure significantly curbed SIRT3 and FOXO3a levels, whereas SBP and TSP treatments reversed this trend and significantly upregulated FOXO3a and SIRT3 expression in a dose-dependent regime.
As it could be seen, there is no much difference between TSP 10 и 100 and no much difference between TSP and SBP. Or maybe better to write, how much is the difference.
4. There is no mention, why do the authors used this cell culture. Is it the appropriate model? Better to introduce the justification into the section “Introduction”
5. The are not any discussion about TSP benefits, for example,
6. ABB not full. There is no HS in ABB and
“PC-12 Rat pheochromocytoma cell” better to change “PC-12 - Rat pheochromocytoma cell line”
PC-12 is a cell line that was derived from a transplantable rat pheochromocytoma. This cell line can be used in neuroscience and toxicology research. https://www.atcc.org/products/crl-1721
7. It would be better to calculate the degree of protection, protection rate,%, as it is described in the published article “Neuroprotective and Memory-Enhancing Effects of Antioxidant Peptide From Walnut (Juglans regia L.) Protein Hydrolysates, Liu M. et al., 2019.
8. In the section "Samples" it is not clear how TSP and SBP have been obtained. SBP have been obtained in the laboratory, and TSP have been bought. So, TSP is not original and it is commercially available? In the previous paper of the authors of this manuscript it have been listed the purification of some peptides - references 14,25. But in this paper this monomeric peptide has not been obtained.
Author Response
Response to Reviewer 2 Comments
Dear honoured reviewers,
We really appreciate you and the reviewers for your comments on our Manuscript: “The SWGEDWGEIW from Soybean Peptides which Reduce Oxidative Damage-mediated Apoptosis in PC-12 cells and Mitochondria by Activating SIRT3/FOXO3a Signaling Pathway” We express our sincere gratitude and thankfulness for your time and precision in reviewing our manuscript. The responses to the comments are as follows. For your kind information, we have carefully dealt with the comments of the reviewers as follows: All the places which we changed have already been marked yellow in our paper, for the purpose of highlight. We hope the revised manuscript meets the standard of publication. Thank you!
Point 1: In the title of the manuscript it’s written “neurons “, but all the experiments have been done only on PC-12 cell line. There were not any neurones used as the object of research, because PC-12 cell line is pheochromocytoma cell line. So, to speculate that “TSP has the potential to replace SBP and reduce neuronal oxidative damage.” may be not correct.
Response 1: Thank you very much for the comments and suggestions. Our work is based on the PC-12 cell, which is often used for preliminary research on the prevention and treatment of neuronal damage. It has already been mentioned in the article that it is possibility, but not certain and these are only preliminary results. In order to prevent misunderstanding, we have removed the neurons in the title and manuscript.
Point 2: The authors write in the “Conclusions” section: that “it is for the first time” but there is some data on this topic. For example, where the soybean peptides were purified and analyzed in detail. [Amakye, W. K., Hou, C., Xie, L., Lin, X., Gou, N., Yuan, E., & Ren, J. (2021). Bioactive anti-aging agents and the identification of new anti-oxidant soybean peptides. Food Bioscience, 42, 101194. doi:10.1016/j.fbio.2021.101194]
Response 2: Thank you very much for the comments and suggestions. We have removed “for the first time” and rewritten the conclusion section (Line297-303).
Point 3: Throughout the paper the authors single peptide (TSP) with the total soybean peptide. The difference is not too big, as it could be seen on the figures. There are not any digital presentation of the difference in percentiles, only visual. For example, as it is displayed in Figure 5A, compared with the NC group, hydrogen peroxide exposure significantly curbed SIRT3 and FOXO3a levels, whereas SBP and TSP treatments reversed this trend and significantly upregulated FOXO3a and SIRT3 expression in a dose-dependent regime. As it could be seen, there is no much difference between TSP 10 и 100 and no much difference between TSP and SBP. Or maybe better to write, how much is the difference.
Response 3: Thank you very much for the comments and suggestions. We have added the difference protection rate% between TSP and SBP, which is marked in yellow color from line194-197,203-204,211-213,260-261.
Point 4: There is no mention, why do the authors used this cell culture. Is it the appropriate model? Better to introduce the justification into the section “Introduction”
Response 4: Thank you very much for the comments and suggestions. We have introduced and explained why PC-12 cell line is used in this article in the introduction section, (Lines 70-76).
Point 5: The are not any discussion about TSP benefits, for example,
Response 5: Thank you very much for the comments and suggestions. We have addressed the advantages of TSP in lines 66-70,276-282.
Point 6: ABB not full. There is no HS in ABB and “PC-12 Rat pheochromocytoma cell” better to change “PC-12 - Rat pheochromocytoma cell line”. PC-12 is a cell line that was derived from a transplantable rat pheochromocytoma. This cell line can be used in neuroscience and toxicology research. https://www.atcc.org/products/crl-1721.
Response 6: Thank you very much for the comments and suggestions. We have added abbreviations in the manuscript and changed the “PC-12 Rat pheochromocytoma cell” to “PC-12 Rat pheochromocytoma cell line” (Lines 304-321).
Point 7: It would be better to calculate the degree of protection, protection rate(%), as it is described in the published article “Neuroprotective and Memory-Enhancing Effects of Antioxidant Peptide From Walnut (Juglans regia L.) Protein Hydrolysates, Liu M. et al., 2019.
Response 7: Thank you very much for the comments and suggestions. We have calculated and added the protection degree and protection rate of peptides in lines 194-197,203-204,211-213,260-261.
Point 8: In the section "Samples" it is not clear how TSP and SBP have been obtained. SBP have been obtained in the laboratory, and TSP have been bought. So, TSP is not original and it is commercially available? In the previous paper of the authors of this manuscript it have been listed the purification of some peptides - references 14,25. But in this paper this monomeric peptide has not been obtained.
Response 8: Because in the previous intervention of peptides on sleep, we found that peptides rich in tryptophan play a very good role in promoting sleep. TSP is the single peptide rich in tryptophan found in our research. Therefore, we synthesized the current TSP through peptide synthesis company. In the "Sample" section, we described the source of TSP and SBP in detail. The reference of our previous paper is Reference 15 but references 14,25 were placed by mistake. Therefore, Reference 14 and 25 have been deleted and replaced with Reference 15 in lines 88-91.
Reviewer 3 Report
Authors of the manuscript evaluated SBP and TSP neuroprotective effect on PC-12 cells against oxidative stress (H2O2) and then mitochondrial dysfunction, activating the SIRRT3-FOXO3a signaling axis.
They further argue that this observable effect of SBP or TSP will go into clinical trials. In the presented study, authors investigated only one cell line PC-12 pheochromocytoma. All these conclusions on the effect of SBO and TSP should be also preceded by in vivo studies. The title of the work suggests that we will find such data in the manuscript. But unfortunately, not.
I think that the authors will either supplement it with data from in vivo experiments or change the journal to a lower-ranked scientific journal. In my opinion there are too little data for publication in Molecules, MDPI.
Author Response
Response to Reviewer 3 Comments
Dear honoured reviewers,
We really appreciate you and the reviewers for your comments on our Manuscript: “The SWGEDWGEIW from Soybean Peptides which Reduce Oxidative Damage-mediated Apoptosis in PC-12 cells and Mitochondria by Activating SIRT3/FOXO3a Signaling Pathway” We express our sincere gratitude and thankfulness for your time and precision in reviewing our manuscript. The responses to the comments are as follows. For your kind information, we have carefully dealt with the comments of the reviewers as follows: All the places which we changed have already been marked yellow in our paper, for the purpose of highlight. We hope the revised manuscript meets the standard of publication. Thank you!
Point 1: Authors of the manuscript evaluated SBP and TSP neuroprotective effect on PC-12 cells against oxidative stress (H2O2) and then mitochondrial dysfunction, activating the SIRRT3-FOXO3a signaling axis. They further argue that this observable effect of SBP or TSP will go into clinical trials. In the presented study, authors investigated only one cell line PC-12 pheochromocytoma. All these conclusions on the effect of SBP and TSP should be also preceded by in vivo studies. The title of the work suggests that we will find such data in the manuscript. But unfortunately, not. I think that the authors will either supplement it with data from in vivo experiments or change the journal to a lower-ranked scientific journal. In my opinion there are too little data for publication in Molecules, MDPI.
Response 1: Thank you very much for your comments and suggestions. PC-12 cells are neurons from pheochromocytoma, a malignant lesion of the adrenal medulla in rats. When cultured in vitro, it can show some growth characteristics similar to neurons, such as cell aggregation and the appearance of fiber ridge. Therefore, these cells show many characteristics and can be used in the study of neurotoxicity, neuroprotection, and neurocognition. The explanation has been given in lines 50-76. At present, we have only conducted a preliminary study on cells, and will carry out an in-depth in vivo study on the mouse in future. In order to avoid ambiguity, we have modified and rewrote this part and hope that experts will give a chance to publish it in this journal.
Round 2
Reviewer 2 Report
- Mitochondria should be deleted from the title, because mitochondrion was not isolated from PC-12 cells.
- Fig.2-4: the designations of the columns above are incomprehensible, it is not explained what are the letters a, b, c or “ab”, “ac” mean, what is the sense. Why does the first column start with the letter c or d?
- Fig.5: it is written in notes: (A) AR-C17 effect on SIRT3 and FOXO3a protein expression. What is the AR-C17? This ABB was not used before (in the section 2.5 it was, but without any explanation).
- The “Conclusion” section contain repeats: “various biological techniques” and in the same sentence |various biological methods”. This section contains too general words, the specific and intriguing results are not given. It is desirable to supplement the conclusion with fundamentally new data obtained in the work to make it sound more scientific and interesting to the readers.
Author Response
Response to Reviewer 2 Comments
Dear honoured reviewers,
We really appreciate you and the reviewers for your comments on our Manuscript: “The SWGEDWGEIW from Soybean Peptides Reduce Oxidative Damage-mediated Apoptosis in PC-12 cells by Activating SIRT3/FOXO3a Signaling Pathway” We express our sincere gratitude and thankfulness for your time and precision in reviewing our manuscript. The responses to the comments are as follows. For your kind information, we have carefully dealt with the comments of the reviewers as follows: All the places which we changed have already been marked yellow in our paper, for the purpose of highlight. We hope the revised manuscript meets the standard of publication. Thank you!
Point 1: Mitochondria should be deleted from the title, because mitochondrion was not isolated from PC-12 cells.
Response 1: Thank you very much for the comments and suggestions. “Mitochondria” hasbeen deleted from the title.
Point 2: Fig.2-4: the designations of the columns above are incomprehensible, it is not explained what are the letters a, b, c or “ab”, “ac” mean, what is the sense. Why does the first column start with the letter c or d?
Response 2: Thank you very much for the comments and suggestions. We used a biological significance labeling method Wherein the highest column in the column analysis graph is designated as “a”, and the column with the significant difference is designated as “b”, and so on. If the difference between a and b is not significantly different (P < 0.05) then it will be marked as “ab” and so on.
Point 3: Fig.5: it is written in notes: (A) AR-C17 effect on SIRT3 and FOXO3a protein expression. What is the AR-C17? This ABB was not used before (in the section 2.5 it was, but without any explanation)..
Response 3: Thanks for your suggestion. The correction has been made and “AR-C17” has been replaced with “SBP or TSP”(line139,296).
Point 4: The “Conclusion” section contain repeats: “various biological techniques” and in the same sentence |various biological methods”. This section contains too general words, the specific and intriguing results are not given. It is desirable to supplement the conclusion with fundamentally new data obtained in the work to make it sound more scientific and interesting to the readers..
Response 4: Thanks for your suggestion. The conclusion section has been checked and modified accordingly (Lines 300-305).

Reviewer 3 Report
The significantly corrected manuscript presents the results in a different (proper) optics. The title has been made more precise and the fragments added to the text complete it. Now this is a brand new paper.
Author Response
Response to Reviewer 3 Comments
Dear honoured reviewers,
We really appreciate you and the reviewers for your comments on our Manuscript: “The SWGEDWGEIW from Soybean Peptides Reduce Oxidative Damage-mediated Apoptosis in PC-12 cells by Activating SIRT3/FOXO3a Signaling Pathway” We express our sincere gratitude and thankfulness for your time and precision in reviewing our manuscript. The responses to the comments are as follows. For your kind information, we have carefully dealt with the comments of the reviewers as follows: All the places which we changed have already been marked yellow in our paper, for the purpose of highlight. We hope the revised manuscript meets the standard of publication. Thank you!
Point 1: The significantly corrected manuscript presents the results in a different (proper) optics. The title has been made more precise and the fragments added to the text complete it. Now this is a brand-new paper.
Response 1: Thank you very much for the comments and suggestions. We have revised the title of the manuscript again to delete “Mitochondria” to make the title expression more accurate.
